# Challenges to Recruiting Men on Active Surveillance for Prostate Cancer in Clinical Chemoprevention Trials

**DOI:** 10.3390/cancers15041257

**Published:** 2023-02-16

**Authors:** Nagi B. Kumar, Saira Bahl, Jasreman Dhillon, Michael Poch, Brandon Manley, Roger Li, Michael Schell, Julio Powsang

**Affiliations:** 1Cancer Epidemiology Program, Moffitt Cancer Center and Research Institute, Tampa, FL 33612, USA; 2Department of Pathology, Moffitt Cancer Center and Research Institute, Tampa, FL 33612, USA; 3Department of Genitourinary Oncology, Moffitt Cancer Center and Research Institute, Tampa, FL 33612, USA; 4Department of Biostatistics, Moffitt Cancer Center and Research Institute, Tampa, FL 33612, USA

**Keywords:** prostate cancer, challenges in clinical trial recruitment, prevention trials

## Abstract

**Simple Summary:**

Patients and public participation in clinical trials is critical to discovering and testing the effectiveness and safety of new drugs to prevent or cure diseases, including cancer. Although it is estimated that >70% of Americans are inclined to participate in clinical trials, less than 5% of adult cancer patients participate in clinical trials. We and others have observed several challenges with recruitment and accrual in clinical trials. The goal of this manuscript is to review our experience in determining protocol and patient level challenges to recruiting prostate cancer patients in clinical cancer chemoprevention trials conducted in a Comprehensive Cancer Center. We report here, contemporary strategies that we have adopted to overcome these challenges to recruit subjects in clinical trials. These strategies can better enable research teams select, focus and invest in strategies that are the most productive and efficient for recruiting target populations to meet recruitment goals.

**Abstract:**

Clinical trials play a critical role in evidence-based medicine, when rigorous scientific methodology is utilized to discover and test the effectiveness and safety of new drugs to prevent or cure diseases, including cancer. Participation in clinical trials thus becomes key to successful completion of these trials. Although it is estimated that >70% of Americans are inclined to participate in clinical trials, less than 5% of adult cancer patients participate in clinical trials. There is thus a large gap between those inclined to participate in clinical trials and actual participation in clinical trials. As with trials targeting men with prostate cancer (PCa) on active surveillance (AS), where the target population is mostly over 50 years of age, others have observed several challenges with recruitment and accrual in clinical trials. The participation rate is currently unavailable for men on primary and secondary chemoprevention trials. Additionally, with unanticipated environmental factors such as a pandemic or other natural emergencies that may severely impact the economy, personal property, travel and person-to person contact for study-related procedures, there is a need to continuously identify these challenges and determine solutions to recruitment barriers in chemoprevention trials to ensure timely completion of early phase trials. Recent studies regarding the impact of the pandemic on clinical trial recruitment have shown that cancer prevention trials were relatively more negatively impacted compared to cancer treatment trials. The goal of this manuscript is to review our experience in continuously evaluating the protocol and patient level challenges to recruiting subjects on AS for PCa in this cancer chemoprevention trial conducted at the Comprehensive Cancer Center (CCC) and report the contemporary strategies that we are utilizing to continue to recruit subjects in this trial. We provide data from our current trial as an example while discussing future strategies to improve overall clinical trial recruitment. These strategies can inform future design of contemporary cancer chemoprevention trials and, additionally, better select, focus and invest in strategies that are the most productive and efficient for recruiting target populations.

## 1. Introduction

The American Cancer Society estimates that there will be 268,490 new cases of prostate cancer (PCa) in the United States (US) in 2022, and 34,500 men will die from this disease [1]. It is reported that approximately 84% of PCas diagnosed are low-grade cancers, where the 5-year survival rate is nearly 100% compared to 31% in men diagnosed with metastatic disease. This is the result for increased screening for PCa using serum prostate-specific antigen (PSA), resulting in a significant increase in the detection of low-grade PCas (Gleason ≤ 6), a disease that has been shown to pose little risk of disease progression or death [2,3,4,5]. On the other hand, overdetection of PCas has been shown to result in overtreatment of low-grade PCa, resulting in increased morbidities with negligible or no benefits for cancer-specific survival [4,6]. Active surveillance (AS) has thus evolved as a recommended management strategy for men with low-grade disease, providing the benefit of an individualized approach to carefully monitoring disease progression using PSA kinetics, imaging studies and periodic biopsies for histologic progression, sufficient to permit timely therapeutic intervention [6]. Using this approach, in a large cohort of men on AS followed over 15 years, Klotz et al. (2015) [7] demonstrated a 98% disease-specific survival for Gleason 3+3 tumors. Other large cohort prospective studies have confirmed the safety and relative effectiveness of this approach [8].

However, several major challenges have been identified in this patient population on AS. Based on the results of the US National Cancer Database (2010–2011) [9], challenges to AS include concerns about undergrading and a significant variation in criteria used to define men at low risk and eligible for AS. For example, of men with PCa, 39.8%, 28.5% and 10.7% were determined to be low risk and eligible for AS by Klotz (least stringent) [10], D’Amico (intermediate stringent) [11] and modified Epstein criteria (most stringent), respectively [12]. Additionally, the percentage of men receiving AS was much lower in the national sample evaluated by Maurice et al. (2015) [9] across all criteria, 6.5%, 7.4% and 12.1%, demonstrating continued overtreatment that has not substantially changed based on the evidence used in the 2012 recommendations [13]. In a large data set, Leyh-Bannurah et al. [14] assessed insignificant prostate cancer (iPCa) rates after robot-assisted radical prostatectomy (RARP) in contemporary patients who were preoperatively eligible for active surveillance (AS). iPCa indicates no risk of PCa progression. Results indicated that greater AS stringency resulted in more AS ineligible patients despite harboring iPCa. As a result, patients, especially older men, were at risk for overtreatment. Other challenges include patient-related factors such as anxiety, depression, doubts about the possible progression of the disease, as well as higher decisional conflict regarding selection of AS [15,16]. Men on AS have thus been reported to ultimately opt for treatment without any major change in tumor characteristics. On the other hand, men on AS are a subgroup who are highly motivated and eager to make positive lifestyle changes to further reduce their risk of PCa progression [16,17,18], providing an opportunity for preventing progression through pharmacologic means [19,20,21]. Taking into consideration these caveats, men on AS are an ideal target for chemoprevention interventions with promising agents, to further reduce progression to later stage disease as well as anxiety during AS. Previous chemoprevention strategies have included large phase III trials with 5-alpha-reductase inhibitors, finasteride and dutasteride [22,23,24], trace element selenomethionine and/or vitamin E, demonstrating greater risk for high-grade disease [25] or no reduction in risk of progression, severely limiting their clinical adoption. Our team and others have evaluated several approaches (WHEL study) [26] and agents (selenium, vitamin E [25], isoflavones [27], lycopene [28] and green tea catechins [29]) targeting prostate carcinogenesis. Among the agents evaluated to date, green tea catechins (GTCs), specifically epigallocatechin gallate (EGCG), a catechin in green tea, appears most promising. Currently, we are recruiting from the target population of men on AS for PCa both at a Comprehensive Cancer Center (ClinicalTrials.gov Identifier: NCT04300855) as well as nationally from the National Clinical Trials Network (ClinicalTrials.gov Identifier: NCT04597359).

Although it is estimated that >70% of Americans are inclined to participate in clinical trials, less than 5% of adult cancer patients participate in clinical trials [30,31,32]. There is thus a large gap between those inclined to participate in clinical trials and actual participation in clinical trials. These data concerning participation rates are currently unavailable for men on primary and secondary chemoprevention trials. As with trials targeting men with PCa on AS, where the target population is mostly over 50 years of age [33], others have observed several challenges with accrual. Additionally, with unanticipated environmental factors such as a pandemic or other natural emergencies such as hurricanes in the State of Florida that may severely impact the economy, personal property, travel and person-to-person contact for study-related procedures, there is a need to continuously identify these challenges and determine solutions for recruitment barriers in chemoprevention trials to ensure timely completion of early phase trials.

The goal of this manuscript is to review our experience in continuously evaluating the protocol and patient level challenges to recruiting subjects on AS for PCa in this cancer chemoprevention trial conducted at the Comprehensive Cancer Center (CCC) and report the contemporary strategies that we are utilizing to continue to recruit subjects in this trial. We provide data from our current trial as an example while discussing future strategies to improve overall clinical trial recruitment. These strategies can inform future design of contemporary cancer chemoprevention trials and additionally, to better select, focus and invest in strategies that are the most productive and efficient for recruiting the target populations.

## 2. Methods

In the following sections, we provide an overview of the cancer chemoprevention clinical trial, proactive measures incorporated in the (a) infrastructure (b) protocol and design of the study procedures; (c) physician-and study-team-related factors; (d) social media, advertisement, mass mailing, posters in clinics, digital marketing of trial; and (e) subject-related factors that were considered. However, with the pandemic within 6 months of activating the trial (and subsequent suspension of the trial to recruitment), we discuss additional challenges and strategies that were incorporated to increase recruitment. Data regarding the number of subjects screened for the trial, the number eligible, the number who agreed to participate and the methods used to recruit the subjects are tabulated. The reason for non-participation in the clinical trial was obtained from the eligible participants when they were willing to provide this.

### 2.1. Overview of the PCa Trial

Overview of Study Design: Our proposed study is a randomized double-blinded clinical trial to evaluate the bioavailability, safety and effectiveness by which a standardized formulation of whole Green Tea Catechin (Sunphenon 90D^®^ (Taiyo International, Inc., Minneapolis, MN, USA)), containing 400 mgs EGCG (BID) vs. placebo, administered for 24 months in a cohort of men on AS for PCa. An IND and an IRB approval will be obtained prior to study initiation. The primary endpoint will be to assess the rate of clinical progression from baseline to end of study at 24 months in men on active surveillance (National Comprehensive Cancer Centers Network (NCCN) guidelines for very low, low and favorable intermediate risk) 63 for PCa (Gleason score (GS) 3+3 OR predominant Gleason grade 2; 7 (3+4), ≤33% of biopsy cores positive for cancer and ≤50% involvement of any core), treated with standardized GTC (400 mg EGCG bid) vs. placebo, with rate of clinical progression defined as a composite outcome on repeat prostate biopsy >33% of biopsy cores positive for cancer or >50% of any biopsy tissue core positive for cancer or adverse reclassification of Gleason sum >3+3 or >3+4, respectively, at the end of study (EOS) biopsy. To complement the primary composite endpoint, other intermediate endpoint biomarkers (IEBs) of the drug effect on (a) PSA and PSA kinetics (PSA, PSA doubling time and PSA density), (b) 17-gene expression panel (Decipher) and (c) incidence of cancer in the post-intervention biopsy between participants treated with GTC and those treated with placebo will be evaluated. Safety will be monitored using NCI common toxicity criteria and serum complete blood count (CBC) and comprehensive metabolic profile (CMP). Compliance to study agent intake will be monitored throughout the trial via pill counts and self-reported daily study-agent intake logs. Adherence to study agent will be monitored by monitoring plasma levels of the catechin EGCG from baseline to end of trial. Patient reported symptoms such as lower urinary tract symptoms (LUTSs) using the LUTS Symptoms Scale and quality of life (QOL), using the Rand Short-Form (SF) 36 will be obtained. This clinical design permits the rigorous evaluation of safety and treatment effects of GTC that will ultimately inform the development of Phase III clinical trials for secondary chemoprevention of low-grade PCa. This study will create a specimen repository with which to test future novel hypotheses to validate molecular targets other than the genomic marker proposed. To assess other genetic or environmental factors that may influence incidence and progression of PCa in men, whole blood, serum, urine and prostate biopsies will be stored in aliquots for analyses as planned and as funding becomes available. Using banked specimens preserved for DNA, RNA and protein analyses, we will be able to assess novel biomarkers and potential molecular targets of GTC.

With our experience in the past, we carefully considered protocol-related and patient-level challenges that can impact participation in this specific clinical trial and proactively developed strategies to overcome these barriers. These strategies were developed based on the feedback from the genitourinary oncologists, urologists, the clinical research coordinators as well as representatives from the target patient population.

### 2.2. Proactive Strategies to Overcome Protocol-Related Challenges

#### 2.2.1. A Recruitment Infrastructure

The project planned involving recruitment of human subjects is a multi-disciplinary endeavor, undertaken by faculty from nutritional sciences, genitourinary oncology (GU), pathology, molecular biology and biostatistics programs at the Moffitt Cancer Center at the Department of Oncologic Sciences. The GU oncology program provides an outstanding infrastructure to conduct chemoprevention trials with a history of successfully recruiting PCa patients at various stages of treatment in clinical trials. The Moffitt Cancer Center Genitourinary Oncology Program aims to (1) elucidate mechanisms of action of crucial molecules in prostate carcinogenesis and tumor progression and investigate their impact on therapeutic efficacy and (2) prospectively assess the clinical utility of the molecules/agents for therapeutic and chemoprevention interventions. The goal of this program is to change the standard of care for people at risk for and with PCa. The research team are faculty members in this program and are involved in the team’s education, research and personalized management of the cancer patients in this program. Subjects are recruited from the Moffitt Cancer Center’s Department of Genitourinary Oncology which has an outstanding and established infrastructure for conducting chemoprevention trials.

#### 2.2.2. Protocol and Design of the Study Procedures

Subjects will be recruited from the Moffitt Cancer Center and from physician practices, referring physicians at each of the affiliate sites to Moffitt. We have successfully utilized these strategies in recruiting subjects in clinical trials by developing a staged, tailored and interactive recruitment process emphasizing communication and relationship building with community-based physicians, oncologists and institutions, providing thorough and complete information sessions to the affiliate and community physicians and support staff, and the public at these locations and trial staff in their clinical offices and institutions were made available for education, training and recruitment purposes. Only referring MDs consenting to comply with allowing patients to be followed at Moffitt Cancer Center will be included in the trial. This infrastructure is critical for patient recruitment and retention.

There are several specific strategies that are used to identify patients who will meet the criteria for specific clinical trials at the Moffitt Cancer Center, specifically PCa trials. These include (a) communication strategies by way of weekly GU Tumor Board team meetings to identify new patients in real time who may be eligible for trial with a diagnosis of PCa with a 3 + 3 or 3 + 4 Gleason score who are placed on active surveillance; (b) patients informed by their treating MD or physician assistant (PA) or nurse practitioner (NP) post biopsy and upon receiving the patient’s pathology results of Gleason score and tumor volume via telephone or Zoom and presenting the opportunity to participate in the trial; (c) the Moffitt Cancer Center has a formal and established active surveillance program, targeting this patient population where over 200 new patients are placed each year on active surveillance. Prostate cancer patients already on active surveillance who are followed in the GU Oncology clinic will be referred to the research team for study information and willingness to participate.

Since hospitals are large and overwhelming for every patient, it is critical to ensure that any burden and anxiety related to the trial is reduced for the participant. Clear instructions in writing must be communicated to the patient prior to their arrival for their initial screening visit. Once the potential subject arrives at the cancer center, the clinical research coordinator CRC meets the potential subject and all procedures, signing of informed consent form (ICF), clinical lab procedures, a brief physical with the treating MDs team and other data collection measures are performed in the same building with CRC helping them navigate through the clinical appointments. A concerted attempt is made to reduce wait time for each of the steps during each visit of the subjects to the study site by ensuring appointments are close together in time and by keeping research and clinical staff up-to-date on patient status.

In this phase II randomized trial, the duration of intervention was 24 months. Although this duration is considered relatively long for phase II clinical trials, we aligned the study visits to be scheduled during the same times that they visited their MDs as a standard of care visit during active surveillance (initial biopsy, follow up visit every 6 months with follow up biopsy at 24 months during AS). One of the other barriers to participation in trials that has been reported is a narrow eligibility criterion [34,35,36,37]. In the elderly population, exclusion criteria are predominantly due to other comorbid conditions [38], especially in cancer trials where a reported 60% of criteria for exclusion were other co-morbidities or performance status. To improve access to trials and ensure generalizability of results of trials as well as to provide a real-world scenario, especially in prevention trials, it is critical to carefully select eligibility criteria to be more inclusive of the target population, without compromising safety. Taking these factors into consideration, we minimized criteria for exclusion due to comorbidities other than cancers and hepatitis B or C. We included subjects with no major organ dysfunction and included periodic safety monitoring relevant to the study agent (CBC CMP and liver function tests (LFTs), Common Terminology Criteria for Adverse Events (CTCAE) monitoring). If patients are unable to come to the research site for safety lab draws, they are permitted to go to their closest clinical lab (Quest labs) for their safety blood draws. Since a vast majority of men over 50 are reported to take a vitamin or mineral supplement, we restricted any supplement containing GTC. All subjects in the trial were also provided a vitamin supplement to discourage use of other supplements. To reduce wait time, study agent supply is mailed via secure mail to the patients upon randomization directly by the investigational pharmacy.

#### 2.2.3. Physician- and Study-Team-Related Factors

We and others have shown that the efforts of experienced and committed physicians’ investigators and research teams who are key stakeholders have been shown to be vital to the success of clinical trial recruitment. Others have also demonstrated that physician decision or preference was the primary reason for non-participation even when a trial was available, and the patient was eligible for a trial [36,37]. Physician’s time restriction due to clinical volume and responsibilities have been cited as the most salient barrier to their lack of referral to trials [39,40,41]. Other factors such as availability of alternate treatments [39,40] have been reported. However, since no other known treatments are currently available for men on active surveillance for low-grade PCa, these factors may not be relevant to this patient population. To offset the time taken to discuss the trial and the informed consent process, after the MD briefly introduces the trial to their patients, the treating MD’s nurse practitioners (NPs) or physician assistants (PAs) present the trial to the potential subjects. Most importantly, the MDs in the study are co-investigators in this clinical trial who are fully committed and who played a key role designing the trial to align with the AS guidelines of the NCCN [42].

##### Formation of a Recruitment and Retention Team

We also formed a Recruitment/Intervention and Retention Team that participated in the initial design of the recruitment procedures. The recruitment team will include the PI and clinical trial coordinators. This team’s role is to ensure implementation of all established procedures consistently for recruitment, intervention and retention of trial subjects. The team will participate in developing the methods for recruitment and retention and will schedule monthly teleconferences to discuss challenges and solutions and successful strategies and revise procedures for recruitment and retention. A monthly screening and recruitment log will be completed and submitted to the PI from each site. Specific data from screening logs include number of biopsies reviewed (pre-screening), potentially eligible subjects, number definitively determined ineligible, number of subjects randomized, anticipated next steps in AS treatment plan for each participant and specific reasons why eligible subjects were not recruited. In addition, a comprehensive review of screening logs, current recruitment strategies, the recruitment and retention team’s meeting minutes and other communications are completed each quarter by the PI to categorize issues as patient- or protocol-related and evaluate other barriers impacting recruitment of subjects in this phase II clinical trial. Barriers are categorized as (a) patient-, (b) protocol- or (c) infrastructure-related. Based on data obtained from the recruitment logs and meeting minutes, protocol-, patient- and infrastructure-related issues are identified and modifications to the protocol to improve recruitment are developed by the administrative team and implemented without compromising the original goals of the research trial.

#### 2.2.4. Social Media, Advertisement, Mass Mailing, Posters in Clinics, Digital Marketing of Trial

Upon receipt of approval from Institutional Review Boards (IRBs), with the assistance of the Institutional public relations and marketing teams, an initial national and local media campaign was initiated using social media, print and web-based communications of the clinical trial. A decision aid brochure was developed to describe the trial rationale, eligibility criteria, responsibility of subjects and a contact number, to be distributed at men’s health events, community medical clinic sites, churches, community organizations and pharmacies. The trial was listed in the ClinicalTrials.gov website of the National Institutes of Health. Supportive marketing and public relations departments at both sites that use community radio, newspaper and other published media advertisements, direct and indirect mailings, internet postings, development and distribution of decision aids and print media exposure of clinical trials, utilizing cultural and literacy competent and experienced teams have demonstrated success in attaining target recruitment numbers in other cancer prevention trials. However, the clinical team was conservative using this approach and, in the past, has been cautious regarding social media advertisements for clinical intervention trials [43,44].

### 2.3. Subject-Related Factors

With respect to subject-related factors, we take into consideration several factors that have presented a challenge in previous trials. Research teams that pay meticulous attention to patient recruitment, retention and frequent monitoring of subject screening, recruitment and randomization logs to inform revisions to trial procedures have been shown to significantly improve recruitment in clinical trials.

Ultimately, the decision to participate in a clinical trial rests with the patient. Patients may consult with family members or close friends who may also influence their decision to participate in a clinical trial [45]. Others are simply motivated with a willingness to be altruistic [46]. Most patients are reported to participate based on wanting to find the best possible treatment for their current disease, especially when there is no alternate treatment other than to be closely monitored by the medical team [47,48]. Most patients report that they do not like the chance of being randomized and that they need to know which arm of the study they were assigned to. Fear of randomization is one of the most commonly reported reasons for non-participation [46]. Others report additional visits to the research site, travel time, costs and completing monitoring records as other reasons for participation in trials [49]. Advocacy groups as well as patients in our previous studies have reported anxiety with information presented in the informed consent of potential side effects as one of the barriers to participation [50].

Based on these barriers, we ensured that the cost of the study-related procedures will be paid by the research and only the standard of care procedure costs will be covered by their insurance carrier. We clarify this issue when we discuss ICF with each subject. The informed consent was written with several team members at various levels reviewing the ICF for simplicity while conforming to the requirements of the Institutional Review Boards. We have also observed a significant burden reported by subjects, especially if the trial involves compliance to agent and diet, frequent visits, completion of monitoring tools and most importantly if these studies include invasive procedures for biomarker evaluation such as biopsy. We have thus aligned our study visits and data collection to the current AS program requirements, and minimized the frequency of visits to the cancer center except for safety monitoring. Telephone contact will be encouraged with calls from the research staff to the subject at least once a month between visits to encourage compliance. We have revised the protocol to provide access to subjects close to their homes to obtain blood draws for safety monitoring when needed, to reduce clinic visits.

Chemoprevention trials that are targeted to high-risk individuals, who already have a level of anxiety, are often unwilling to be randomized to placebo-control arms of the trial and “want” to be in the intervention arm. In addition, most of the agents under study are available in similar doses as over the counter (OTC) supplements or in “natural” forms that subjects have access to. This increased availability has encouraged subjects to opt out of participation in randomized clinical trials, where there is a chance of being assigned to a placebo arm. To provide the opportunity for more men to be randomized to the treatment arm, we designed the study for subjects to be randomized 2:1 to receive the GTC with 400mgs EGCG BID (experimental arm) or placebo.

### 2.4. Pandemic-Related Barriers

With increasing cases of COVID-19 and to control the pandemic, Moffitt Cancer Center placed a hold on most research activities, including this clinical trial as it involved in-person visits. Recruitment to the trial was thus halted. The safety risks for in-person visits were unknown and potentially high, especially for a cancer patient population, where the additional risk was not known. Patients over the age 65 were considered to be at exceptional risk based on evolving data and several comorbidities were added to potential risk for severe disease and even death. The trials were revised to evaluate the number of visits to the clinic. Once the vaccinations as well as booster doses were available, the cancer center offered free vaccinations to all patients and spouses. Although the trial initiated recruitment in August 2020, patients on AS for PCa did not want to come in to volunteer for a clinical trial or for any study visits. The timepoints when subjects were not willing to come in followed the COVID-19 waves between 1 March to 25 April 2020, followed by the months of October 2020 to January 2021.

### 2.5. Natural Disaster-Related Factors

So far in 2022, the State of Florida has had two major hurricanes, including Ian, which made landfall on the Gulf Coast of Florida as a Category 4 in October 2022 and Nicole that made landfall in November 2022. The patient catchment area for our Cancer Center encompassing regions affected by these natural disasters, including loss of property, homes as well as clinical sites closing at the Cancer Center, impacted recruitment in the clinical trial.

## 3. Results

Upon receiving initial funding approval from the National Cancer Institute (NCI), the protocol was approved by the Moffitt Scientific Review Committee (SRC) and Advara Institutional Review Board (IRB). We submitted and received approval for the Investigational New Drug (IND) from Food and Drug Administration (FDA) for the study agent and placebo (FDA IND# 143615 (Kumar NB). The trial was registered in ClinicalTrials.gov Identifier: NCT04300855. The accrual of study patients was for the trial initiated on August 21, 2020. A total of 201 subjects were initially screened for the trial (Table 1). However, upon further review, 51 men were no longer eligible. Among those who were no longer eligible, some subjects were deceased (2); there was reported progression of disease (2) and an increase in serum PSA (22); other treatments were initiated (radiation, hormone therapy) (15); some moved out of State (9); and some were diagnosed with another cancer other than prostate (1). Fifteen of these subjects were recruited and 39 subjects of AS continue to be interested and are currently undergoing the screening process. Additionally, 17 subjects did not respond to phone calls or emails. Fifty-four declined to participate. Among those who declined to participate, reasons for non-participation are shown in Table 2. Of the 25 subjects who were randomized for trial, a 16% attrition rate was observed, with 4 subjects dropping out of study for various reasons (1 moved out of state, 1 underwent disease progression and for 2 the trial was too difficult or they were unable to travel).

Although the reasons for not wanting to participate were collected and assessed to discern trends to see if revisions to the protocol were needed, we did not observe any specific trend other than those who were not interested (16/54) and were unable to provide a specific reason for not participating. Although these numbers are still similar to those observed in other trials, our team still has continued to observe residual anxiety related to the pandemic. Although we have used various strategies for recruiting subjects in long-term clinical trials such as advertising including utilizing social media, we find that the number of subjects who are recruited using these strategies for trials targeting early disease and duration of intervention (1–2 years) is less than 1%. Similar to our previous experience [51], we received no response from any other strategies such as social media, advertisement, mass mailing, posters in clinics, digital marketing of trial. The most efficient and successful recruitment strategies continue to be those where the initial study presentation is completed by their MD/PA/NP after which the patient is referred to the study senior CRC for further screening and recruitment procedures. All subjects (100%) recruited to the trial and those referred and pending were recruited successfully initially with the presentation of the trial to the potential subject by the MD/PA/NP.

## 4. Discussion

In a recent cohort study, examining the experience of cancer clinical trial enrollment 1 year after the COVID-19 outbreak, Unger et al. demonstrated that clinical trial enrollments decreased during the full year of the COVID-19 pandemic, primarily for cancer control and prevention trials [52]. On the other hand, there was no strong evidence of enrollment reductions in treatment trials. Based on several clinical trials, Unger et al. [30] provide a model pathway based on classifying these factors into (a) structural (clinic access, assessment of trial availability); (b) clinical (assessment of patient eligibility to clinical trials); (c) attitudinal (discussion of trial with physician and trial participation offered/not offered); and (d) attitudinal (patient) (patient decision). We [51] have previously broadly classified these factors into (a) protocol-related factors (design of the trial) and (b) patient-related factors [51]. In this report, we initially classified the barriers as follows: (I) protocol-related challenges: (a) recruitment infrastructure; (b) protocol and design of the study procedures; (c) physician- and study-team-related factors; (d) social media, advertisement, mass mailing, posters in clinics, digital marketing of trial; (II) subject-related factors; (III) pandemic-related factors; and (IV) natural disaster-related factors. This enabled our team to consider these specific challenges and proactively develop strategies in the planning of the trial. However, the challenges due to the pandemic and natural disasters were unanticipated, emphasizing an urgent need to continuously evaluate these challenges and to develop contemporary strategies that may improve trial recruitment. The data from the study clearly demonstrate that although strategies to recruit subjects from previous experiences and published data were proactively employed in the design of this clinical trial, the COVID-19 pandemic had a major impact on the PCa trial recruitment that caused the need for further refinement of the trial. All trial data provided involved post COVID-19 pandemic vaccination availability and shortly after the second variant that impacted that State of Florida and the US. This experience clearly demonstrated a need to continuously evaluate the protocol, patient and other environmental challenges and to develop contemporary strategies to recruit subjects in clinical trials.

The trial was eventually opened for recruitment in August 2020. However, like the experience reported nationally for prevention trials, subjects continued to not want to participate in the trial, especially since the trial was targeting men on AS, who are not on active treatment. The pandemic challenged the conventional strategies used to present clinical trials to potential subjects, face to face. Since clinics were operating with reduced capacity and reduced in-person visits, recruiters were required to work virtually. Although other interventions (exercise, yoga) were administered via telemedicine [53,54], chemoprevention trials with a study drug under IND were not conducive to telemedicine interventions. With consistent evidence that clinical trials that enroll patients at proposed rates are critical to ensure internal validity and improve cancer mortality and morbidity, our goal is to recruit subjects in clinical trials efficiently and achieve our recruitment goals in a timely manner.

To offset the time lost due to the pandemic to recruit subjects in this trial, we began the process of recruiting subjects from additional cancer center sites including the George Washington University Cancer Center, Washington DC, and the University of Kansas Cancer Center, Westwood, Kansas, with enthusiastic PI and research teams with expertise in recruiting PCa patients in clinical trials. The number of eligible men who can be recruited from these sites is estimated at 25 annually. Additionally, we have now created a Recruitment and Retention team, including the two other sites that have been added to the trial. The protocol has been amended to reflect the addition of the two sites. Procedures for drug dispensing to these sites, obtaining the generic multivitamin, monitoring, data safety and other regulatory processes have been established. Moffitt Cancer Center’s External Site Coordination (ESC) team will be working with the PI and the external sites to ensure that the study is running smoothly at the external sites. We have revised the protocol, ICF and IRB and FDA approval has been received for these changes.

## 5. Conclusions and Future Directions

As with our previous report [51], the role of the treating MD and the clinical team is critical to the success of clinical trial recruitment in cancer prevention trials. Our experience with clinical trial recruitment demonstrated that 100% of the subjects recruited in the clinical trial were referred to the trial by their MD/MDs, PA or NP. Additionally, future clinical trials should continue to proactively take into consideration infrastructure-, protocol-, research team- and patient-related challenges when designing intervention trials in prostate and other cancer patient populations. Additionally, the research team should continue to maintain records of reasons why subjects refuse to participate in clinical trials and evaluate the need to revise study design or other factors that can mitigate or remove these barriers to participation. It is clear from our experience during the pandemic that there is a need for flexibility and swift changes to the protocols for implementation to continue to conduct cancer prevention trials while maintaining the scientific integrity of the cancer prevention clinical trials [55]. Artificial intelligence [56] techniques for identifying patients, telemedicine for patient communication during trials and re-evaluation of the protocol for the need for frequent face-to-face clinic visits for blood draws and biopsies without compromising the outcomes of trials and safety are a few strategies that have been proposed. Although frequent amendments to protocols may delay implementation, strengthening relationships with SRCs/IRBs and FDA prior to amending the protocol and implementation may expedite and enhance the conduct of cancer prevention clinical trials in the future.

## Figures and Tables

**Table 1 cancers-15-01257-t001:** Summary of eligible subjects contacted.

Summary	As of 11/10/2022
Recruited and Randomized	25
Additional Enrolled/Scheduled	12
Patient has agreed and awaiting follow up biopsy/MRI in the next 6 months	3
On AS–CRC will contact patient based on AS follow up schedule	39
Left voicemails or sent emails	17
Declined	54
No longer eligible	51
**Total contacted**	**201**

**Table 2 cancers-15-01257-t002:** Reasons for declining trial.

Reason for Declining Participation	Number
Things are going well/does not see benefit	5
Not comfortable with end of study biopsy	3
Concerned about experimental drug, anti-drugs, placebo	2
Does not want to discontinue green tea supplements or not limit green tea drinking	2
Does not have time to commit/long duration of study	8
Distance (e.g., 2 patients live far away and cannot drive)	4
Currently seeing too many doctors, too many appointments	2
Illness (Parkinson’s disease; gastrointestinal issues; cirrhosis; thyroid and kidney disease; wife in hospice, cancer)	7
Not interested (looked over protocol and no interest)	5
No specific reason	16
Participated in another trial	1
**Total**	**54**

## Data Availability

Data regarding this clinical trial can be found at Clinicaltrials.gov/NCT04300855.

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
