# Peer review of "Challenges to Recruiting Men on Active Surveillance for Prostate Cancer in Clinical Chemoprevention Trials"

_cancers, 2023, doi:10.3390/cancers15041257_

Round 1

Reviewer 1 Report

The authors showed the difficulty in recruiting men on active surveillance for prostate cancer in clinical chemoprevention trials. The manuscript is informative. However, it needs several revisions.

1.     The primary and secondary endpoints of this study should be discussed. If the primary endpoint is overall survival, the study needs very long duration to evaluate the results. Therefore, another surrogate endpoint should be used in this study.

2.     The other limitation of this study is the difficulty in study continuation. The patients should use the chemopreventive reagents for a long duration. That might be a reason for the low rate of recruitment. It should be discussed.

Author Response

Please see attached responses.

Reviewer 2 Report

Dear Authors,

I read with interest your manuscript entitled: Challenges to Recruiting Men on Active Surveillance for Prostate Cancer in Clinical Chemoprevention Trials”.

Overall, the paper is interesting and well written.

Just a couple of limitations:

·      Abstract is not interesting as the manuscript, please try your best to improve it

·      Tables are not easily readable, please try your best to modify it

Finally, please consider citing the following pertinent paper in introduction section:

·      Leyh-Bannurah SR, Wagner C, Schuette A, Addali M, Lia- kos N, Urbanova K, et al. The impact of age on pathological insignificant prostate cancer rates in contemporary robot-as- sisted prostatectomy patients despite active surveillance eligi- bility. Minerva Urol Nephrol 2022;74:437–44. 

Author Response

please see responses in the attached document.
